# SynHING: Synthetic Heterogeneous Information Network Generation for Graph Learning and Explanation

## Abstract

Graph Neural Networks (GNNs) have achieved remarkable success on relational data, yet their interpretability in heterogeneous information networks (HINs) remains underexplored, largely due to the absence of reliable benchmarks with ground-truth explanations. We introduce **SynHING**, a synthetic HIN generation framework that supports both graph learning and explainability research. SynHING constructs synthetic graphs by extracting motifs from reference networks, assembling them through motif-guided composition, and refining them via post-pruning to preserve structural and statistical fidelity. Importantly, SynHING is *not limited to the extracted motifs*: users can incorporate their own motifs of interest, which the framework integrates seamlessly into the generated graphs. This flexibility enables controlled and reproducible studies across diverse domains. Experiments on IMDB, Recipe, ACM, and DBLP demonstrate that SynHING produces realistic and semantically consistent HINs, while providing a principled testbed for evaluating Heterogeneous GNNs (HGNNs) and explanation methods. To our knowledge, SynHING is the first framework to enable user-defined, motif-aware HIN synthesis, establishing a foundation for interpretable and reproducible research in heterogeneous graph learning.

## 1 Introduction

Graph Neural Networks (GNNs) have advanced state-of-the-art performance across diverse graph-based tasks, including community detection, molecular property prediction, and recommendation systems (Shchur & Günnemann, 2019; Stokes et al., 2020; Cui et al., 2020). Among them, *heterogeneous information networks* (HINs)—graphs with multiple node and edge types—offer a rich modeling paradigm for complex relational data. This has spurred rapid progress in heterogeneous GNNs (HGNNs), exemplified by HAN (Wang et al., 2019a), MAGNN (Fu et al., 2020), and transformer-based variants (Yun et al., 2020; Hu et al., 2020b).

Alongside predictive performance, there is a growing demand to interpret GNN decision-making. For homogeneous graphs, synthetic benchmarks with planted structures and ground-truth explanations have been instrumental in advancing explainability (Dwivedi et al., 2020; Abbe, 2017; Ying et al., 2019). In contrast, heterogeneous settings lack comparable resources. Existing graph generators often rely on random wiring or oversimplified templates, producing graphs that are semantically inconsistent and ill-suited for benchmarking. Moreover, most approaches constrain the structural patterns that can be generated, limiting flexibility for systematic explainability studies.

We propose **SynHING**, a synthetic HIN generation framework designed to support both graph learning and explainability research. SynHING builds synthetic graphs by extracting motifs from a reference HIN, assembling them through motif-guided composition, and refining the results with post-pruning to preserve structural and statistical fidelity. Unlike prior approaches, SynHING is *not limited to the extracted motifs*: users can specify motifs of interest, which the framework integrates seamlessly into the generated graphs. This design makes SynHING broadly adaptable across domains and tasks, while enabling controlled and reproducible evaluation of HGNNs and explanation methods.

We validate SynHING on four widely used HINs: IMDB[1], Recipe (Majumder et al., 2019), ACM (Wang et al., 2019a), and DBLP[2], spanning entertainment, e-commerce, and scholarly communication. Experiments show that SynHING generates semantically coherent and structurally realistic graphs, serving as robust testbeds for both HGNN models and explanation techniques. To our knowledge, SynHING is the first framework to enable controlled, user-defined motif integration in HIN synthesis, providing a foundation for interpretable and reproducible research in heterogeneous graph learning.[3]

## 2    Related Works

### 2.1    Synthetic Graph Generation

Synthetic data has long been central to machine learning, spanning images (Kingma & Welling, 2013; Frid-Adar et al., 2018), tabular data (Bowyer et al., 2011; Xu et al., 2019; Figueira & Vaz, 2022), and music (Dong et al., 2018). With the rise of GNNs, synthetic graphs have become equally important for testing robustness and generalization. Classical generators such as the Stochastic Block Model (SBM) (Snijders & Nowicki, 1997), Degree-Corrected SBM (DC-SBM) (Abbe, 2017), and GraphWorld (Palowitch et al., 2022) primarily focus on homogeneous graphs, often neglecting semantic heterogeneity. While tools like GNNExplainer (Ying et al., 2019) are often validated on simple synthetic motifs, no existing generator creates HINs with semantic structure and controllable explanatory units. SynHING fills this gap by enabling the generation of semantically consistent HINs with user-defined motifs that support systematic explainability studies.

### 2.2    Explanation Techniques for Graph Neural Networks

Explainability is key to building trustworthy machine learning systems. For GNNs, existing methods can be grouped into two categories: (i) *inherently interpretable models*, such as ProtGNN (Dai & Wang, 2021), which embed explanation mechanisms directly into the model; and (ii) *post-hoc explainers*, which identify important subgraphs (Luo et al., 2020; Yuan et al., 2021) or features (Ying et al., 2019) after training. Despite rapid progress, most evaluations still rely on homogeneous synthetic graphs with overly simplistic motifs (e.g., grids, houses) (Ying et al., 2019), limiting their ability to capture the complexity of real-world settings. Recent work has begun to explore explainability in heterogeneous contexts (Li et al., 2023; Lv et al., 2023), but the absence of principled synthetic HINs with controllable motifs has prevented rigorous benchmarking. SynHING directly addresses this gap by offering a flexible testbed tailored for heterogeneous explainability research.

### 2.3    Graph Datasets with Ground-Truth Explanations

Graph explanation methods are commonly evaluated on molecular datasets, such as MUTAG (Debnath et al., 1991), where functional groups serve as ground-truth explanations, or on synthetic benchmarks using simple generative models (e.g., Barabási–Albert, trees) (Ying et al., 2019). While useful, these datasets are homogeneous and structurally simple, failing to represent the semantic richness and type diversity of real-world heterogeneous graphs. In contrast, SynHING provides a framework that not only extracts motifs from real HINs but also allows users to define additional motifs of interest. This enables synthetic graphs that remain faithful to the structural and semantic distributions of the source data while supporting controlled, motif-level ground truths for explainability evaluation.

## 3    Proposed Method: SynHING

### 3.1    Preliminaries

HINs, also called heterogeneous graphs, consist of multiple node and edge types, which can be defined as a graph $G = (\mathcal{V}, \mathcal{E}, \Phi, \Psi)$, where $\mathcal{V}$ and $\mathcal{E}$ are the set of nodes and edges, respectively.

---

[1]https://www.kaggle.com/datasets/karrrimba/movie-metadatacsv

[2]http://web.cs.ucla.edu/~yzsun/data/

[3]Code will be released upon acceptance.

Each node $v \in \mathcal{V}$ has a type $\Phi(v) \in \mathcal{T}_{\mathcal{V}}$, and each edge $e \in \mathcal{E}$ has a type $\Psi(e) \in \mathcal{T}_{\mathcal{E}}$, where $\mathcal{T}_{\mathcal{V}}$ and $\mathcal{T}_{\mathcal{E}}$ are collections of node and edge types. The node feature matrix is denoted by $F_\phi \in \mathbb{R}^{|\mathcal{V}^\phi| \times d_\phi}$, where $\mathcal{V}^\phi$ is the set of node with node type $\phi$, i.e. $\mathcal{V}^\phi = \{ v \in \mathcal{V} \mid \Phi(v) = \phi \}$, and $d_\phi$ is the feature dimension of the node type $\phi$. Target nodes of the graph $G$, denoted by $\mathcal{V}^{\phi_0}$, are associated with labels collected as $Y \in \mathcal{Y}^{|\mathcal{V}^{\phi_0}|}$, where $\phi_0 \in \mathcal{T}_{\mathcal{V}}$ denotes the target node type.

## 3.2 OVERVIEW OF SYNHING

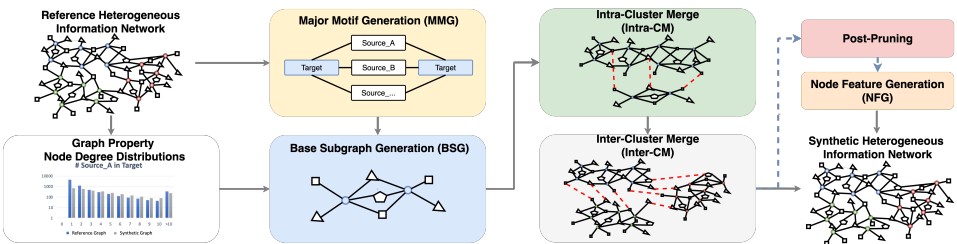

Figure 1: SynHING

We propose SynHING, a framework for generating synthetic HINs $\tilde{G}$ with ground-truth explanations, designed to replicate the structural and statistical properties of a target graph $\hat{G}$. SynHING follows a structured bottom-up pipeline: (1) generating key motifs and base subgraphs, (2) merging subgraphs within (Intra-Cluster) and across (Inter-Cluster) clusters, and (3) applying node feature generation and post-pruning. An overview is shown in Figure 1, with further methodological details presented in the following sections.

## 3.3 MAJOR MOTIF GENERATION (MMG)

Motifs capture semantically meaningful interaction patterns in heterogeneous graphs (e.g., author–paper–venue in DBLP, actor–movie–director in IMDB) and are widely regarded as natural explanatory units. From the perspective of HGNNs, motifs define the relational structures over which message passing and reasoning occur (Wang et al., 2019b), making them suitable ground-truth explanations. However, existing graph generators do not synthesize HINs with such verifiable explanatory structures.

To address this limitation, we design the *Major Motif Generation (MMG)* module, which identifies dominant, explanation-relevant patterns from real networks. These motifs form the building blocks for controlled synthesis and serve as benchmark units for evaluating GNN explainability. Concretely, MMG discovers meta-paths that start and end at target nodes while traversing intermediate node types. A major motif is then constructed by selecting two anchor nodes of the target type and connecting them through all valid meta-paths within a user-defined hop limit (Wang et al., 2019a; Fu et al., 2020). The hop limit can be manually set or aligned with the receptive field of the downstream HGNN (i.e., its $n$-hop neighborhood).

Figure 1 illustrates this process: MMG composes a motif from three one-hop meta-paths, yielding graphlets similar to those observed in real networks (Milo et al., 2002). For example, in IMDB we recover the $G20$ motif, which connects two anchors via bridging nodes, a structure also leveraged in MEGNN (Chang et al., 2022). In addition, MMG is not limited to extraction from existing graphs: researchers can also *define custom motifs*, which the framework integrates seamlessly into the generation process. This flexibility enables SynHING to support both realistic replication of observed patterns and task-specific benchmarking tailored to user needs.

## 3.4 BASE SUBGRAPH GENERATION (BSG)

We generate base subgraphs from major motifs by introducing controlled variations and structural noise. As illustrated in Figure 1, randomness is injected into each motif, which is then augmented with several non-target nodes, or minor nodes. These minor nodes are connected to the target nodes, mimicking real-world graph noise while maintaining core structures.

Minor nodes fulfill two key roles: they help match the degree distribution of the target nodes in the subgraphs to that of the reference graph, modeled as $P^\phi(k)$, where $k$ denotes the number of connections to nodes of type $\phi$, and they serve as junction points for merging operations between subgraphs.

Each base subgraph is created by assigning the same label to the two target nodes within a motif, forming a labeled instance $(S_i, y_i)$, where $S_i = (\mathcal{V}_i, \mathcal{E}_i)$ represents the subgraph structure and $y_i \in \mathcal{Y}$ is its class label. The subgraphs are grouped into sets $\mathcal{K}_y$ for each class $y \in \mathcal{Y}$, forming the foundational components for assembling full graphs.

### 3.5 MERGE TO GENERATE HINS

Conventional methods for graph construction typically involve adding edges between nodes or subgraphs to create a connected graph. However, this approach can lead to the introduction of illegal connections, making it challenging to maintain type-specific constraints and degree distributions in heterogeneous graphs.

To overcome these limitations, we introduce a novel *Merge* operation that fuses two nodes into a single node while preserving all edges from the original pair. This technique maintains local structure, semantic validity, and degree profiles of the reference graph, allowing for the controlled assembly of larger graphs that are consistent with both motif-level semantics and global statistics. Formally, given a graph $G$ and two nodes $v_1, v_2 \in \mathcal{V}$, the merge of $v_2$ into $v_1$ is defined as:

$$(\mathcal{V}', \mathcal{E}') = \text{Merge}(v_1, v_2; G) \tag{1}$$
$$= (\mathcal{V} \setminus \{v_2\}, \mathcal{E} \cup \{(v_1, u) \mid u \in N(v_2), u \neq v_1\} \setminus$$
$$\{(v_2, u) \mid u \in N(v_2)\}), \tag{2}$$

where $(v_1, u)$ denoted the edge connecting node $v_1$ and $u$, and $N(v)$ is the set of neighbors of the node $v$. This operation connects all neighbors of $v_2$ to $v_1$ (except $v_1$ itself), then removes $v_2$ and its edges. We also extend this definition to multiple node pairs by writing $\text{Merge}(\mathfrak{P}; G)$, where $\mathfrak{P} \subseteq \mathcal{V} \times \mathcal{V}$ is a set of node pairs to merge. The order of merges in $\mathfrak{P}$ does not affect the final result. For merging of pairs across multiple graphs $G_1, G_2, \ldots$, we denote $\text{Merge}(\mathfrak{P}; G_1 \oplus G_2 \oplus \ldots)$, where $\oplus$ is the disjoint union operator. Using this operation, we construct the full synthetic HIN through a bottom-up process of Intra- and Inter-Cluster Merges.

#### 3.5.1 INTRA-CLUSTER MERGE (INTRA-CM)

Intra-CM iteratively merges base subgraphs with identical labels to form a cluster $C_y = (\mathcal{V}_y, \mathcal{E}_y)$, emulating the "Superstar" phenomenon observed in community networks (Albert & Barabási, 2002; Abbe, 2017), where certain nodes accumulate many connections and emerge as influential hubs or opinion leaders.

For each class $y \in \mathcal{Y}$, let $\mathcal{K}_y$ denote the set of base subgraphs from the BSG step. These subgraphs are sequentially merged using the *Merge* operator to form cluster $C_y$, and the process is repeated $|\mathcal{Y}|$ times to construct the full set of clusters $C_y \mid y \in \mathcal{Y}$. Because HINs involve multiple node types, Intra-CM is applied separately for each type $\phi$ to ensure semantic consistency across the heterogeneous network. Specifically, the initial subgraph $S_0$ is selected from $\mathcal{K}_y$ to initialize the cluster, denoted as $C_y^0$. At each iteration $i$, a subgraph $S_i = (\mathcal{V}_i, \mathcal{E}_i)$ is chosen from the remaining set $\mathcal{K}_y \setminus \{S_0, \ldots, Si-1\}$, and merged with the current cluster $C_y^{i-1}$ to produce the updated cluster $C_y^i$. The number of *Merge* operation, denoted by $n_{\text{intra}}^\phi$, for each minor node type $\phi \neq \phi_0$ is sampled from a binomial distribution:

$$n_{\text{intra}}^\phi \sim \text{B}\left(n = |\mathcal{V}_i^\phi|, p = p^\phi\right), \tag{3}$$

where $p^\phi$ controls intra-cluster density: larger $p^\phi$ increases connectivity and reduces inter-cluster overlap. To perform merging between nodes in $S_i$ and $C_y^{i-1}$, we define the candidate pair sampling space:

$$\mathfrak{M}_{\text{intra}}^\phi = \{\{v_y, v_i\} \mid v_y \in \mathcal{V}_y^\phi, v_i \in \mathcal{V}_i^\phi\}. \tag{4}$$

This formulation ensures type-specific, semantically valid merges during cluster construction. Subsequently, $n_{\text{intra}}^\phi$ pairs are then sampled uniformly from $\mathfrak{M}_{\text{intra}}^\phi$ into $\mathfrak{P}^\phi \subseteq \mathfrak{M}_{\text{intra}}^\phi$ without replacement.

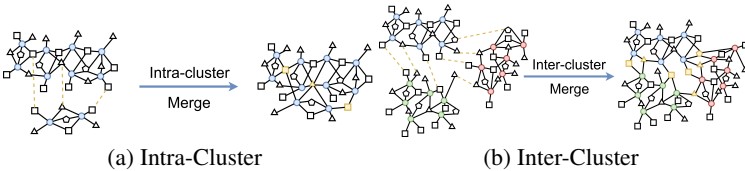

(a) Intra-Cluster             (b) Inter-Cluster

Figure 2: Intra-Cluster and Inter-Cluster Merges

The *Merge* operation is performed on the sampled pairs of all minor node types to merge the $S_i$ into the cluster:

$$C_y^i = \text{Merge}\left( \bigcup_{\phi \in \mathcal{T}_\mathcal{V}, \phi \neq \phi_0} \mathfrak{P}^\phi; \quad C_y^{i-1} \oplus S_i \right), \tag{5}$$

where $\bigcup$ denotes the union over all minor node types. Through this process, we generate label-specific clusters that preserve type constraints while reflecting heterogeneous community structures (Figure 2a).

### 3.5.2 INTER-CLUSTER MERGE (INTER-CM)

Inter-CM combines clusters with distinct labels $\{C_y \mid y \in \mathcal{Y}\}$ from the Intra-CM step to create the final synthetic heterogeneous information network (HIN) $\tilde{G}$. In contrast to the Intra-CM process, this merging occurs concurrently, as the clusters are formed independently without any hierarchical dependencies. Merges are conducted separately for each node type $\phi$ to ensure semantic coherence. The candidate set of cross-cluster node pairs is defined as follows:

$$\mathfrak{M}_{\text{inter}}^\phi = \left\{ \{v_1, v_2\} \mid v_1 \in \mathcal{V}_{y_1}^\phi, \, v_2 \in \mathcal{V}_{y_2}^\phi, \, \{y_1, y_2\} \subseteq \mathcal{Y}, \, y_1 \neq y_2 \right\}, \tag{6}$$

where $\mathcal{V}_{y_1}^\phi, \mathcal{V}_{y_2}^\phi$ are nodes of type $\phi$ from two different clusters $C_{y_1}, C_{y_2}$, respectively. The number of inter-cluster merges ($n_{\text{inter}}^\phi$) is sampled as:

$$n_{\text{inter}}^\phi \sim B\left( n = \sum_{y \in \mathcal{Y}} |\mathcal{V}_y^\phi|, k = q^\phi \right), \tag{7}$$

where $q^\phi$ is the merge probability. A higher value of $q^\phi$ increases cross-cluster merging and reduces cluster separation. The sampled pairs $\mathfrak{P}^\phi \subseteq \mathfrak{M}_{\text{inter}}^\phi$ are then merged to form a complete graph $\tilde{G}$:

$$\tilde{G} = \text{Merge}\left( \bigcup_{\phi \in \mathcal{T}_\mathcal{V}'} \mathfrak{P}^\phi; \quad \bigoplus_{y \in \mathcal{Y}} C_y \right), \tag{8}$$

where $\bigoplus$ denotes disjoint union. For multi-label graphs, merging operations are allowed across all node types, including the target type $\phi_0$, i.e., $\mathcal{T}_\mathcal{V}' = \mathcal{T}_\mathcal{V}$. In contrast, for single-label generation, target nodes are excluded to preserve label integrity ($\mathcal{T}_\mathcal{V}' = \mathcal{T}_\mathcal{V} \setminus \{\phi_0\}$). Following the Inter-CM process, we obtain a fully connected synthetic HIN with controllable structural entanglement, governed by intra- and inter-cluster probabilities (Figure 2b). These parameters allow researchers to study GNN behavior under varying levels of structural entanglement and label separation.

### 3.6 NODE FEATURE GENERATION (NFG)

Following previous research (Palowitch et al., 2022; Tsitsulin et al., 2022), NFG generates features for target nodes by sampling from multivariate normal distributions within clusters, specifically $\mathcal{N}(\mu_y, \alpha)$, where the cluster center $\mu_y$ is drawn from a global distribution $\mathcal{N}(0, \beta)$. The feature signal-to-noise ratio (SNR) is defined as the ratio of inter-cluster distance to intra-cluster covariance, which is controlled by the parameters $\beta$ and $\alpha$. For multi-label target nodes, features are derived from a joint distribution that captures overlapping label semantics. For minor nodes that may have missing features, we use approximations by assigning features based on node IDs, types, or simple encodings (Lv et al., 2021). This strategy ensures semantic coherence and compatibility with lightweight node representations for subsequent models.

### 3.7 POST-PRUNING (P-P)

P-P is an optional step that enables users to manually adjust the distributions of node types to better reflect real-world constraints. For example, in the IMDB dataset, each movie is typically linked to no more than three actors. During the P-P process, node degrees are restricted by predefined thresholds, leading to the removal of excess edges. This edge removal is conducted in a priority-aware manner, ensuring that edges within significant motifs are preserved. This approach maintains the integrity of embedded explanations and guarantees the reliability of evaluation tasks.

### 3.8 COMPLEXITY AND SCALABILITY OF SYNHING

To assess the scalability of SynHING, we conducted a complexity analysis, examining each module theoretically. The processes of MMG and BSG are independent, leading to a time complexity of $O(N)$ for both. The complexity of Intra-CM is $O(N|\mathcal{V}_i| + N|\mathcal{E}_i|)$ or $O(N)$, since $|\mathcal{V}_i|$ and $|\mathcal{E}_i|$ represent the number of nodes and edges in the base subgraph, which remain constant with respect to $N$. The processes in Inter-CM are analogous to those in Intra-CM, resulting in a similar time complexity of $O(N)$. Therefore, the overall time complexity of SynHING is dictated by the number of motifs $N$, resulting in a time complexity of $O(N)$, which highlights the scalability of the framework. A comprehensive theoretical analysis can be found in Appendix A, with empirical runtime data in Appendix B.

## 4 EXPERIMENTAL SETTINGS

**Datasets and HGNNs.** We evaluate the SynHING using synthetic graphs derived from four widely used HIN node classification datasets: IMDB[1], Recipe (Majumder et al., 2019), ACM (Wang et al., 2019a), and DBLP[2]. To identify major motifs, we anchor two target nodes and connect them via feasible meta-paths within a fixed hop limit — two hops for IMDB, Recipe, and ACM, and four hops for DBLP (see Figure 3). For node classification, we use transductive learning, with 24% of target nodes for training, 6% for validation, and 70% for testing, as suggested in (Lv et al., 2021). To validate the synthetic HINs, we employ three prominent HGNNs: HGT (Hu et al., 2020b), SimpleHGN (Lv et al., 2021), and TreeXGNN (Hong et al., 2023) as the encoders (Appendix D and E). Additional details, including studies of minor node degrees of reference graphs, are provided in Appendix G.

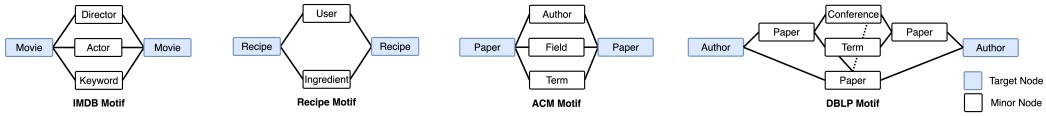

Figure 3: Graph schema and major motifs of the four heterogeneous graph datasets

## 5 RESULTS AND DISCUSSION

Our work focuses on generating synthetic datasets for model evaluation. To the best of our knowledge, no existing approach provides synthetic HINs with explanation ground truths or suitable benchmarks. Therefore, we designed a series of experiments for systematic validation and multi-faceted analysis.

### 5.1 CLUSTER EXCLUSION CONTROLS ENABLE STRUCTURED BENCHMARKING OF HGNNS

To showcase SynHING's capability in generating synthetic HINs with varying levels of cluster exclusion, we utilize the intra-cluster merge probability $p$ and inter-cluster merge probability $q$. These parameters significantly impact the structural purity of the generated graph: higher values of $p$ combined with lower values of $q$ result in stronger intra-cluster connectivity and weaker inter-cluster overlap, leading to more distinctly separated clusters. Thanks to the flexible design of SynHING, this level of exclusion can be easily adjusted. To assess the impact of exclusion levels on model

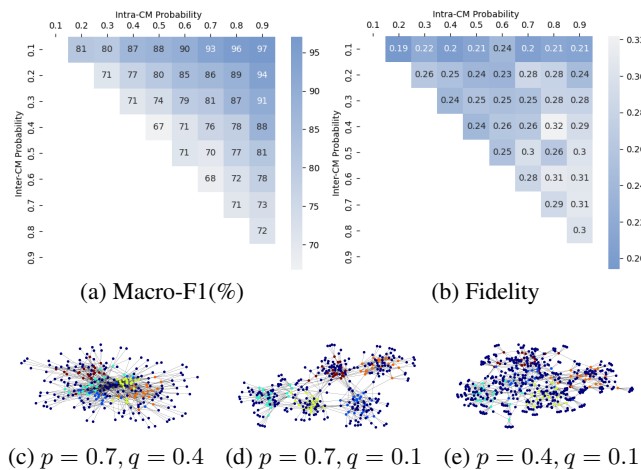

(a) Macro-F1(%)  (b) Fidelity

(c) $p = 0.7, q = 0.4$  (d) $p = 0.7, q = 0.1$  (e) $p = 0.4, q = 0.1$

Figure 4: Visualization of synthetic IMDB in different Intra-/Inter-CM probabilities. Dark blue represents minor nodes. Others indicate target nodes with different labels.

performance, we benchmarked the HGT using synthetic IMDB datasets (Syn-IMDB) created under various combinations of $p$ and $q$, with $p > q$. As shown in Figure 4a, the Macro-F1 score of HGT improves with increasing $p$, indicating better cluster purity, while higher $q$ leads to increased inter-cluster noise and diminished performance. We also visualize the generated Syn-IMDB graphs under three configurations in Figures 4c, 4d, and 4e, highlighting the structural changes driven by the values of $p$ and $q$. The results confirm that SynHING enables precise control over cluster exclusion, aiding in the benchmarking of HGNN models. Similar trends were observed with SimpleHGN and TreeXGNN, with detailed results in Figure 7 in Appendix F.

## 5.2 FIDELITY TRENDS REVEAL THE EXPLANATORY POWER OF MAJOR MOTIFS

To verify that the motifs in synthetic HINs encode essential structural patterns and yield reliable explanatory signals, we evaluate explanation quality using *fidelity-*. This metric measures the average drop in predicted probability when only the features highlighted by the explanation are retained (Ying et al., 2019; Yuan et al., 2020). As *fidelity-* captures the sufficiency of an explanation, it is well aligned with our setting, where SynHING generates explanations directly rather than approximating them. A detailed formula for *fidelity* can be found in Appendix C

Figure 4b shows that the overall fidelity trend is consistent with HGT's Macro-F1 scores: synthetic HINs with greater cluster exclusion achieve lower fidelity scores, indicating higher explanatory sufficiency. Notably, fidelity remains stable across different values of the Intra-CM probability $p$ (Figure 5a). While larger $p$ introduces additional connections beyond the motifs, the core predictive signals continue to revolve around motif structures, confirming their central role in GNN predictions. In contrast, Figure 5b demonstrates that increasing the Inter-CM probability $q$ raises fidelity scores, reflecting degraded sufficiency due to reduced cluster exclusion and the introduction of inter-cluster noise. Finally, Figure 5c highlights the effect of feature quality: higher SNR leads to lower fidelity scores, indicating that informative node features strengthen the sufficiency of motif-based explanations when graph structure is fixed.

Overall, these findings affirm that SynHING's motifs provide reliable ground-truth explanations that capture essential graph patterns. At the same time, they underscore the importance of transparent evaluation practices, as inappropriate settings (e.g., excessive inter-cluster merging) may reduce fidelity and obscure the role of motifs in model decision-making.

## 5.3 ABLATION STUDIES

To assess the contribution of each module within SynHING, we conducted ablation studies by sequentially removing key modules in the IMDB dataset. More specifically, the Random-Motifs study

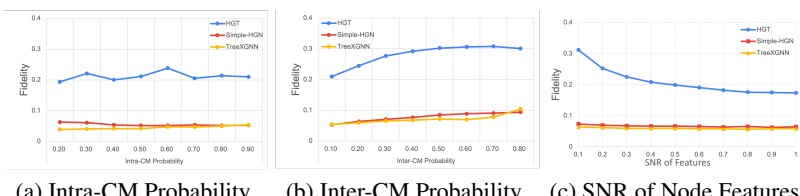

(a) Intra-CM Probability     (b) Inter-CM Probability     (c) SNR of Node Features

Figure 5: Fidelity of major motifs under different SynHING parameter settings

Table 1: Ablation studies of SynHING. Significance levels are indicated as follows: * for $p < 0.05$, ** for $p < 0.01$, and *** for $p < 0.001$, compared to SynIMDB.

|  | SynIMDB | | Random-Motifs | | Random-Merge | |
|---|---|---|---|---|---|---|
|  | Macro-F1 (%) | Micro-F1 (%) | Macro-F1 (%) | Micro-F1 (%) | Macro-F1 (%) | Micro-F1 (%) |
| HAN | 82.37 ± 0.45 | 82.42 ± 0.52 | 77.21 ± 0.69*** | 77.20 ± 0.81*** | 37.31 ± 5.57*** | 41.47 ± 4.40*** |
| HGT | 87.86 ± 0.30 | 87.88 ± 0.31 | 80.99 ± 0.55*** | 80.97 ± 0.58*** | 68.52 ± 3.74*** | 69.11 ± 3.24*** |
| SimpleHGN | 87.60 ± 0.49 | 87.66 ± 0.49 | 83.04 ± 0.48*** | 83.02 ± 0.48*** | 72.77 ± 6.33*** | 72.88 ± 6.30*** |
| TreeXGNN | 87.68 ± 0.35 | 87.70 ± 0.36 | 81.77 ± 1.14*** | 81.73 ± 1.15*** | 68.92 ± 6.57*** | 69.07 ± 6.56*** |

involves disabling the MMG module and randomly generating motifs. The Random-Merge study focuses on randomly merging nodes without performing Intra-/CM and Inter-CM operations. As shown in Table 1, the performance of Random-Motifs declines markedly compared to the original SynIMDB across all HGNNs—HAN, HGT, SimpleHGN, and TreeXGNN—with reductions of -5.16%, -6.87%, -4.56%, and -5.91% in Macro-F1, respectively. This decrease underscores the importance of the MMG module. Additionally, the performance of Random-Merge further diminishes, with losses of -45.06%, -19.34%, -14.83%, and -18.76% in Macro-F1 for HAN, HGT, SimpleHGN, and TreeXGNN, respectively. These results reveal consistent trends across the HGNNs and illustrate the efficacy of the proposed Merge method for generating synthetic HINs.

## 5.4 SYNTHETIC GRAPH PRETRAINING LEADS TO POSITIVE TRANSFER IN REAL HIN TASKS

To investigate whether structurally similar synthetic graphs can enhance downstream learning, we perform transfer learning experiments by pretraining on synthetic graphs and then finetuning on their corresponding real-world counterparts. This approach is inspired by previous research indicating that transfer learning without semantic alignment can lead to negative transfer (Hu et al., 2020a; Rosenstein et al., 2005). We assess similarity from two perspectives: (i) When the synthetic and reference graphs are both structurally and semantically aligned, we anticipate positive transfer. (ii) Conversely, if the synthetic graphs are intentionally corrupted, we expect to see negative transfer, which would impair model performance. Detailed implementation information is provided in Appendix H.

**Positive Transfer.** We compare models finetuned on real graphs with and without pretraining on their synthetic counterparts. Table 2 shows that pretraining on SynHING-generated HINs significantly improves performance across four datasets. For instance, on IMDB, Macro-F1 improves by up to 3% for HGT and 2% for SimpleHGN. We also observe reduced standard deviations in IMDB, Recipe, and ACM, indicating greater model stability. These consistent gains across both HGNNs confirm that SynHING graphs support effective and reliable transfer.

Table 2: Performance comparison: Evaluating pretraining on Syn-HINs vs. no pretraining, followed by fine-tuning on real-world graphs. Boldface highlights improvements. Significance levels are indicated as follows: * for $p < 0.05$, ** for $p < 0.01$, and *** for $p < 0.001$.

|  |  | HGT | | SimpleHGN | |
|---|---|---|---|---|---|
| Dataset | Pretrained on | Macro-F1 | Micro-F1 | Macro-F1 | Micro-F1 |
| IMDB | - | 63.00 ± 1.19 | 67.20 ± 0.57 | 63.53 ± 1.36 | 67.36 ± 0.57 |
|  | **Syn-IMDB** | **66.10 ± 0.21*** | **68.03 ± 0.53*** | **65.52 ± 0.50*** | **68.45 ± 0.53*** |
| Recipe | - | 57.26 ± 1.84 | 56.98 ± 2.02 | 60.29 ± 1.31 | 60.15 ± 1.41 |
|  | **Syn-Recipe** | **57.82 ± 0.46** | **57.83 ± 0.64** | **60.40 ± 0.22** | **60.21 ± 0.23** |
| ACM | - | 91.12 ± 0.76 | 91.00 ± 0.76 | 93.42 ± 0.44 | 93.35 ± 0.45 |
|  | **Syn-ACM** | **92.55 ± 0.20*** | **92.54 ± 0.21*** | **94.16 ± 0.43*** | **94.11 ± 0.44*** |
| DBLP | - | 93.01 ± 0.23 | 93.49 ± 0.25 | 94.01 ± 0.24 | 94.46 ± 0.22 |
|  | **Syn-DBLP** | **93.88 ± 0.25*** | **94.35 ± 0.23*** | **94.27 ± 0.58** | **94.73 ± 0.56** |

**Negative Transfer.** To simulate negative transfer, we construct malicious synthetic graphs by: (i) Node shuffling: Randomly permuting rows in the adjacency matrix $A$ to disrupt graph structure and homophily. (ii) Feature shuffling: Randomly permuting rows of the feature matrix $F$, breaking the correspondence between node features and labels. Table 3 reveals that manipulations significantly degrade performance, confirming negative transfer. Feature shuffling generally has a more severe impact, due to the disruption of input-label alignment and impairing message passing.

Table 3: HGT performance comparison: Pretraining on node- and feature-shuffled Syn-HINs, then fine-tuning on real HINs. Boldface highlights the lowest score, and underline marks the second-lowest. Significance levels are indicated as follows: * for $p < 0.05$, ** for $p < 0.01$, and *** for $p < 0.001$.

| | Pretrain on SynHING | Macro-F1 | Micro-F1 |
|---|---|---|---|
| IMDB | w/o Shuffled | $66.10 \pm 0.21$ | $68.03 \pm 0.53$ |
| | Node Shuffled | $64.54 \pm 0.58$*** | $67.44 \pm 0.59$ |
| | **Feature Shuffled** | **$62.06 \pm 1.28$***** | **$63.96 \pm 0.79$***** |
| Recipe | w/o Shuffled | $57.82 \pm 0.46$ | $57.83 \pm 0.64$ |
| | **Node Shuffled** | **$47.87 \pm 0.83$***** | **$47.66 \pm 0.88$***** |
| | Feature Shuffled | $55.46 \pm 1.09$** | $55.55 \pm 1.11$** |
| ACM | w/o Shuffled | $92.55 \pm 0.20$ | $92.54 \pm 0.21$ |
| | Node Shuffled | $90.45 \pm 0.49$*** | $90.45 \pm 0.48$*** |
| | **Feature Shuffled** | **$89.02 \pm 1.54$***** | **$89.09 \pm 1.46$***** |
| DBLP | w/o Shuffled | $93.88 \pm 0.25$ | $94.35 \pm 0.23$ |
| | Node Shuffled | $93.56 \pm 0.32$ | $94.06 \pm 0.30$ |
| | **Feature Shuffled** | **$93.25 \pm 0.29$**** | **$93.75 \pm 0.30$**** |

It is important to note that when the goal is to study explainable ground truths, synthetic graphs need not resemble any real dataset. SynHING supports the generation of novel HINs via user-defined major motifs, enabling controlled experiments beyond real-world constraints. To quantitatively assess structural similarity, we also apply the Comparing Degree Distribution (CDD) metric (Darabi et al., 2023). Results confirm that SynHING allows controlled generation of synthetic HINs with high fidelity to real-world structural patterns. Further details are provided in Appendix G.

### 5.5 SynHING Supports the Evaluation of HGNN Explanation Methods

We applied our generated synthetic datasets to HGNN explainers and demonstrated that our synthetic datasets can indeed be used for HGNN explanation algorithms. Further details can be found in Appendix I. However, research on heterogeneous GNN explainers remains limited, highlighting the need for further exploration in this area.

## 6 Conclusion

We presented **SynHING**, a general framework for generating synthetic HINs with controllable explanatory structures. By extracting motifs from real HINs and assembling them via Intra-/Inter-CM operations, SynHING produces large-scale graphs that preserve both structural fidelity and semantic consistency. Crucially, the framework is not restricted to extracted motifs: users can also define their own motifs of interest, enabling flexible and reproducible evaluation across diverse domains. Experiments on IMDB, Recipe, ACM, and DBLP demonstrate that SynHING generates realistic and semantically coherent graphs, while providing ground-truth motifs that serve as reliable benchmarks for explanation methods. To our knowledge, SynHING is the first framework to support user-defined, motif-aware HIN synthesis, addressing a fundamental gap in explainability research for heterogeneous graphs. We hope SynHING will serve as a foundation for advancing interpretable, reproducible, and domain-adaptive research in heterogeneous graph learning.

### Reproducibility

To ensure reproducibility, the source code and datasets are included in the supplementary materials.

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

APPENDIX

# Table of Contents

## A   SYNHING'S COMPLEXITY AND SCALABILITY

In this section, we theoretically analyze the complexity of the SynHING framework module by module to demonstrate its scalability. Let $N$ represent the motif number, determining the scale of the generated graph. We demonstrate that the total time complexity for SynHING is $O(N)$. For simplicity, we omit node type in the analysis for both Intra-CM and Inter-CM, as nodes merge only with those of the same type, making the complexity linear to the number of types. The generations of the motif (MMG) and the base subgraph (BSG) can be parallelized, and execution time is linear to the number of items. Therefore, the time complexity of MMG and BSG is $O(N)$.

The complexity of Intra-CM is analyzed step-by-step as follows:

(i) Eq.(3), we determine $n_{\text{intra}}$, the number of pairs to be sampled.

(ii) Eq.(4), we sample $n_{\text{intra}}$ nodes from $\mathcal{V}_y$ and $\mathcal{V}_i$, and pairing them as $\mathfrak{P}_{\text{intra}}$.

(iii) Eq.(5), the merge process.

(iv) We offset the "incoming" subgraph $S_i$ by the maximum IDs of $C_y$ (graph disjoint union).

(v) We drop the selected nodes in $\mathcal{V}_i$.

(vi) We re-index the edges in $\mathcal{E}_i$ based on the mapping determined by $\mathfrak{P}_{\text{intra}}$.

The complexity of steps (i), (ii), and (v) is $O(|\mathcal{V}_i|)$. The complexity of step (iv) is $O(|\mathcal{V}_i| + |\mathcal{E}_i|)$. The complexity of step (vi) is $O(|\mathcal{E}_i|)$. One iteration complexity is $O(|\mathcal{V}_i| + |\mathcal{E}_i|)$. There will be $(N - |\mathcal{Y}|)$ iterations, making the total complexity $O(N|\mathcal{V}_i| + N|\mathcal{E}_i|)$ or $O(N)$, as $|\mathcal{V}_i|$ and $|\mathcal{E}_i|$ are the number of nodes and edges in the base subgraph, which are constant w.r.t. $N$.

Following a similar process, the complexity of Inter-CM is analyzed:

(i) Eq.(6) and Eq.(7), we identify all $\binom{|\mathcal{Y}|}{2}$ combinations of clusters and determine the number of pairs that need to be merged for each combination.

(ii) After the pair number has been determined, we derive the node number that needs to be merged for each cluster. We randomly select nodes from each cluster based on this number without replacement. (iii) Merge process eq.(8).

(iii) We offset all the clusters $C_y$. (the graph disjoint union in eq.(8)).

(iv) We drop one of the nodes in each pair in $\mathfrak{P}_{\text{inter}}$ in $\mathcal{V}_y$ for each cluster.

(v) We re-index the edges in $\mathcal{E}_y$ for each cluster based on the mapping determined by $\mathfrak{P}_{\text{inter}}$.

The complexity of steps (iv), (v), and (vi) are $O(\sum_{y\in\mathcal{Y}}(|\mathcal{V}_y| + |E_y|))$, $O(\sum_{y\in\mathcal{Y}}|\mathcal{V}_y|)$, and $O(\sum_{\mathcal{Y}}|\mathcal{E}_y|)$. Since $\sum_{y\in\mathcal{Y}}|\mathcal{V}_y| \leq N|\mathcal{V}_i|, \sum_{y\in\mathcal{Y}}|\mathcal{E}_y| \leq N|\mathcal{E}_i|$. The complexity of Inter-CM is $O(N|\mathcal{V}_i| + N|\mathcal{E}_i|) = O(N)$.

Overall, SynHING can generate large-scale HINs in a reasonable timeframe, with complexity of $O(N)$ where $N$ denotes the number of motifs.

## B    SYNHING'S EMPRICIAL RUNTIME

To validate the theoretical linear time complexity of SynHING in Appendix A, we conducted scaling experiments on the SynIMDB dataset by varying the number of motifs and measuring the runtime of the core merge process.

We specifically measured only the merge stages, excluding Major Motif Generation (MMG) and Base Subgraph Generation (BSG), as both are clearly linear in complexity—each subgraph is processed independently in these stages.

Table 4: Merge process runtime over varying motif counts.

| # Motifs | IntraCM (ms) | InterCM (ms) | Total (ms) |
|---|---|---|---|
| 20 | 5 | 8 | 13 |
| 200 | 43 | 20 | 63 |
| 2,000 | 408 | 129 | 539 |
| 20,000 | 4,116 | 1,230 | 5,347 |
| 200,000 | 40,878 | 13,678 | 54,556 |
| 2,000,000 | 417,085 | 161,362 | 578,447 |

Experiments were run on a machine with an Intel(R) Core(TM) i7-10700 CPU using a single process, with no hardware acceleration or parallelism. As shown in Table 4, the runtime grows linearly with the number of motifs. SynHING successfully processed up to 2 million motifs in under 10 minutes. Note that the IntraCM step can be parallelized across clusters, potentially achieving up to $|\mathcal{Y}|$-fold speedup.

## C    EVALUATION METRICS

We use Micro-F1 and Macro-F1 as evaluation metrics for node classification and fidelity for explanation evaluation. Micro-F1 scoring assesses a model's predictions across all samples, with a tendency to emphasize the majority category. In contrast, Macro-F1 scoring equally weights each category, promoting a balanced evaluation of data across different categories. Therefore, we mainly use Macro-F1 as the major evaluation metric (Wang et al., 2019a; Lv et al., 2021; Hong et al., 2023).

Fidelity is a metric commonly used to evaluate the performance of the explanation model (Yuan et al., 2021; Li et al., 2022). It measures how closely related the explanations are to the model's predictions. If the critical information is included in the explanation subgraph, the classification model prediction probability should be close to the original prediction, resulting in low fidelity. We use fidelity as the evaluation metric to support that the major motifs can be excellent explanations of ground truths. The following are the details of the fidelity score:

$$Fidelity = \frac{1}{N}\sum_{i=1}^{N}\frac{1}{L}\sum_{l=1}^{L}\left\|f(G_i)_{y_l} - f(\hat{G}_i)_{y_l}\right\|, \tag{9}$$

where $f(G_i)_{y_l}$ and $f(\hat{G}_i)_{y_l}$ denote the prediction probability of $y_l$ of the original graph $G_i$ and major motifs $\hat{G}_i$ (explanation subgraph), respectively. We denote $N$ as the total number of target node samples and $L$ as the number of node labels.

## D    BENCHMARK HETEROGENEOUS GRAPH NEURAL NETWORKS

We used three HGNN models, each representing a different underlying concept, to validate the synthetic graphs. The model parameters follow the recommendations of the original paper. The

Table 5: Performance comparison of three HGNNs on real and synthetic HINs

| | IMDB | | Recipe | | ACM | | DBLP | |
|---|---|---|---|---|---|---|---|---|
| | Macro-F1 (%) | Micro-F1 (%) | Macro-F1 (%) | Micro-F1 (%) | Macro-F1 (%) | Micro-F1 (%) | Macro-F1 (%) | Micro-F1 (%) |
| HGT | 63.00 ± 1.19 | 67.20 ± 0.57 | 57.26 ± 1.84 | 56.98 ± 2.02 | 91.12 ± 0.76 | 91.00 ± 0.76 | 93.01 ± 0.23 | 93.49 ± 0.25 |
| SimpleHGN | 63.53 ± 1.36 | 67.36 ± 0.57 | 60.29 ± 1.31 | 60.15 ± 1.41 | 93.42 ± 0.44 | 93.35 ± 0.45 | 94.01 ± 0.24 | 94.46 ± 0.22 |
| TreeXGNN | 65.59 ± 0.89 | 69.28 ± 0.64 | 59.99 ± 0.94 | 59.97 ± 0.96 | 94.32 ± 0.54 | 94.29 ± 0.54 | 94.94 ± 0.63 | 95.24 ± 0.59 |
| | SynIMDB | | SynRecipe | | SynACM | | SynDBLP | |
| | Macro-F1 (%) | Micro-F1 (%) | Macro-F1 (%) | Micro-F1 (%) | Macro-F1 (%) | Micro-F1 (%) | Macro-F1 (%) | Micro-F1 (%) |
| HGT | 87.86 ± 0.30 | 87.88 ± 0.31 | 87.95 ± 1.90 | 87.95 ± 1.88 | 99.45 ± 0.30 | 99.46 ± 0.30 | 97.97 ± 0.75 | 97.99 ± 0.74 |
| SimpleHGN | 87.60 ± 0.49 | 87.66 ± 0.49 | 87.82 ± 0.23 | 87.83 ± 0.23 | 99.41 ± 0.37 | 99.41 ± 0.37 | 98.48 ± 0.41 | 98.48 ± 0.41 |
| TreeXGNN | 87.68 ± 0.35 | 87.70 ± 0.36 | 86.68 ± 0.37 | 86.72 ± 0.36 | 99.12 ± 0.67 | 99.12 ± 0.66 | 99.18 ± 0.18 | 99.18 ± 0.18 |

following briefly introduces the models: (1) HGT (Hu et al., 2020b) adopts a transformer-based design for handling different node and edge types without manually defining the meta-path for the HGNN model. (2) SimpleHGN (Lv et al., 2021) introduces the attention mechanism, projects different node-type features to the shared feature space, and then uses GAT as the HGNN backbone. (3) TreeXGNN (Hong et al., 2023) leverages the decision tree-based model XGBoost to enhance the node feature extraction, assisting the HGNN model in getting more prosperous and meaningful information.

In order to evaluate the performance of SynHING, we utilize the transductive learning approach for node classification tasks and randomly select 24% of the target nodes for training, 6% for validation, and 70% for testing (Wang et al., 2019a; Lv et al., 2021; Hong et al., 2023). We repeated all experiments five times and evaluated performance using average Micro-F1 and Macro-F1 for node prediction and fidelity for interpretation evaluation.

# E  SYNTHETIC HINS WITH GROUND-TRUTH EXPLANATIONS

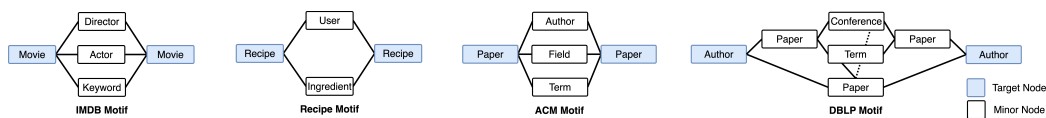

Figure 6: Major motifs of the four heterogeneous graph datasets

We evaluate SynHING using three HGNNs on four synthetic HINs (with Syn- in front) based on their corresponding real-world graphs, shown in Table 5. HGNNs achieve better performance on Macro-F1 and Micro-F1 scores for learning and inference on synthetic graphs compared to real graphs. These improvements can be attributed to the designated major motifs in synthetic graphs, shown in Figure 6, which provide ground-truth explanations for assessing explainability methods and result in synthetic graphs containing purer information for graph learning. We mimic the graph properties of the reference graph and identify the parameters for generating the synthetic graph. This selection ensures that the resulting synthetic graphs closely approximate the graph structure of the referenced graphs. In addition, the degree of exclusion in SynHING can be customized for different motifs and datasets, which will be discussed in the next subsection.

# F  MORE EXPERIMENTAL RESULTS

As found in Figure 4a, a similar trend is found in SimpleHGN and TreeXGNN as shown in Figure 7. It demonstrated that we can use $p$ and $q$ to control the exclusion of clusters within the generating synthetic HIN and benchmark the ability of HGNN graph learning. This allows us to control graph generation with high flexibility.

Figure 8a illustrates the performance changes of HGT, Simple-HGN, and TreeXGNN at different SNRs of the features. It shows that as the SNR increases, the disparity between node features in different groups widens, and it is easier to discriminate between different clusters only based on their features. Consequently, when the classification model makes predictions, it can leverage this additional information in the nodes, leading to improved performance in classification tasks.

We also explored the impact of adjusting the number of major motifs shown in Figure 8b, which directly affects the number of target nodes and the size of the synthetic graph dataset. It is important

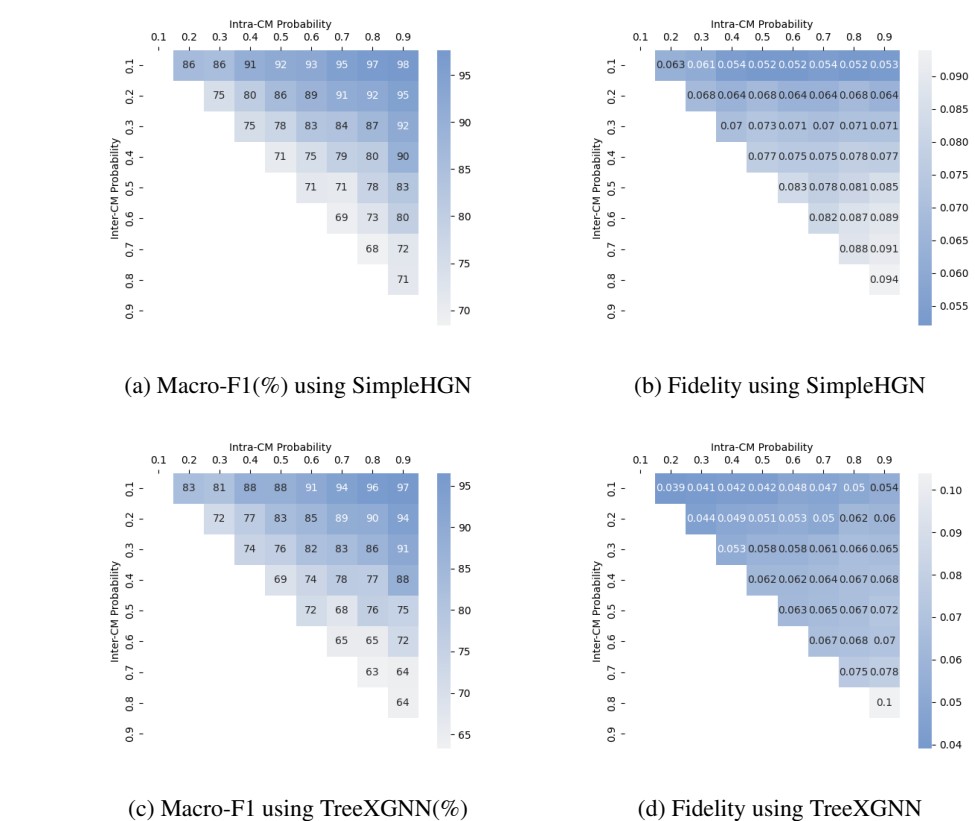

(a) Macro-F1(%) using SimpleHGN

(b) Fidelity using SimpleHGN

(c) Macro-F1 using TreeXGNN(%)

(d) Fidelity using TreeXGNN

Figure 7: Macro-F1 and Fidelity of synthetic IMDB in different Intra-/Inter-Cluster probabilities across different HGNNs

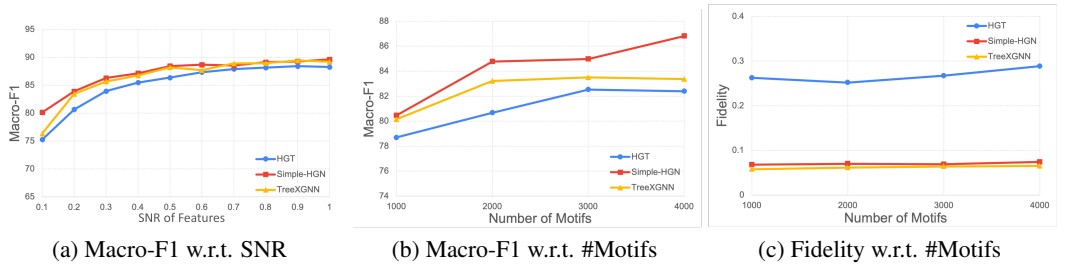

(a) Macro-F1 w.r.t. SNR

(b) Macro-F1 w.r.t. #Motifs

(c) Fidelity w.r.t. #Motifs

Figure 8: Macro-F1 and Fidelity of synthetic IMDB in different SNR and number of motifs

to note that since we kept the hyperparameter settings of the classification model consistent with the original values, rather than finetuning them for each synthetic graph dataset, reducing the dataset size to half caused the model to become overfitted, resulting in a decline in performance.

The fidelity results of HGT, Simple-HGN, and TreeXGNN for varying numbers of major motifs are shown in Figure 8c. When adjusting the number of motifs, which corresponds to the size of the graph, the fidelity performance remains stable.

## G  APPROXIMATING REFERENCED GRAPH

Users can customize the synthetic graph for various scenarios using the parameters of SynHING. including the number of major motifs $N$, the number of clusters $|\mathcal{Y}|$, the Intra-CM probabilities

$p^\phi$, the Inter-CM probabilities $q^\phi$, and the signal-to-noise ratio (SNR) of features $\alpha/\beta$. For example, adjusting the Intra-CM probability $p^\phi$ and the Inter-CM probability $q^\phi$ results in changes in the exclusion of clusters and the difficulty of the synthetic graph. However, these parameters can also be directly determined by the referenced graph $\hat{G}$. Although some statistical properties and network schema have been used for generating graphs, it is further demonstrated that the synthetic graph can approximate the referenced graph more closely by adjusting these parameters: The number of major motifs $N$ can be set as half of the number of target nodes in $\hat{G}$, i.e., $N = \frac{1}{2}|\hat{\mathcal{V}}^{\phi_0}|$, since each motif contains exactly two target nodes. The number of clusters can be determined by the number of labels $|\hat{\mathcal{Y}}|$ in $\hat{G}$. The SNR of features $\alpha/\beta$ can adjust the difficulty of the task on $\tilde{G}$, or users can determine the means and variances of clusters of features by maximum likelihood estimation.

The Intra-/Inter-CM probabilities $p^\phi, q^\phi$ for minor node type $\phi \neq \phi_0$ control the exclusion of clusters, the degree distributions of source nodes, and their counts in the resulting graph $\tilde{G}$. For instance, in Figure 9, we observe the node degree distributions for minor node types in both real-world IMDB and SynIMDB, with $p = 0.7$ and $q = 0.3$. In contrast, Figure 10 compares these distributions with SynIMDB using different probabilities: $p = 0.9, q = 0.8$, and $p = 0.2, q = 0.1$. As depicted, improper selection of $p$ and $q$ can lead to notable deviations in the degree distribution of minor node types.

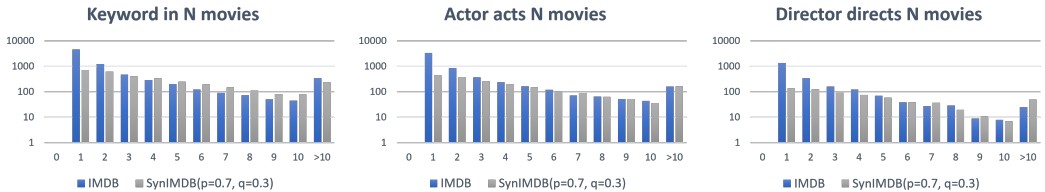

Figure 9: Degree distributions of minor node types in IMDB and SynIMDB

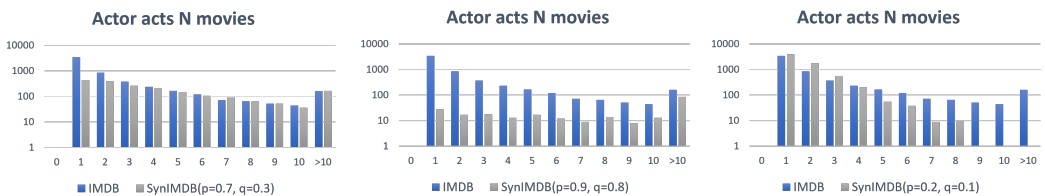

Figure 10: Comparison of degree distribution deviations in SynIMDB with varying Intra-/Inter-CM probabilities

We further applied a statistical-based method, Comparing Degree Distribution (CDD) (Darabi et al., 2023), to measure the structure similarity between real and synthetic graphs. The CDD value ranges between 0 and 1, with 1 indicating that the distribution of the two structures is the same. We applied the settings as Figure 9 and Figure 10 for structure similarity analysis. Table 6 indicates that the generated SynIMDB can be controlled by the Intra-CM/Inter-CM ratio that influences the similarity with real IMDB. When p=0.7 and q=0.3, Macro-CDD and Micro-CDD are 0.8545 and 0.8279, respectively, which is the most similar to the real IMDB compared to the other two settings. This result highlights the effectiveness of SynHING in regulating the generation of synthetic HINs.

Table 6: Comparing degree distribution (CDD) between IMDB and SynIMDB with varying Intra-/Inter-CM probabilities

| Intra-CM (p), Inter-CM (q) | Macro-CDD | Micro-CDD |
|---|---|---|
| p=0.7, q=0.3 | 0.8545 | 0.8279 |
| p=0.9, q=0.8 | 0.7636 | 0.7612 |
| p=0.2, q=0.1 | 0.7597 | 0.7354 |

## H  PRETRAINING AND FINETUNING

In this study, we employ synthetic graphs for pretraining. Models are pretrained based on the recommended settings from their respective original papers, with early stopping applied after 30 epochs without validation set improvement. For finetuning, the weights of the HGNN backbone, excluding the adapter layer that maps the heterogeneous features into shared space, are inherited from the pre-trained model. We note that the weights of the backbone and adapter are trained using different learning rates, as the results are sensitive to the learning rate. For instance, while finetuning from pretrained weights, a lower learning rate for the backbone and a higher learning rate for the adapter generally yield better results, whereas a higher learning rate for the backbone and a lower learning rate for the adapter generally leads to better performance when learning from scratch. Consequently, we conduct a grid search for learning rates in both scenarios, as presented in Tables 2 and 3. For the learning rate of the backbone, we try values of $\{10^{-3}, 10^{-4}\}$. For that of the adapter, we try values of $\{1, 5\} \times \{10^{-2}, 10^{-3}, 10^{-4}\}$.

## I  IMPLEMENTED ON HGNN EXPLAINER

We utilized synthetic ACM and DBLP datasets and input them into the xPath framework (Li et al., 2023). The synthetic IMDB dataset is excluded from this experiment as the corresponding dataset will be a multiple-choice dataset, which is not supported by xPath. The recipe dataset is not used in xPath, so it will not be discussed here. We utilized xPath's default parameters, including the HGNN encoder and explainer. We followed the instructions in xPath, which involved two main steps: (1) Training the HGNN and (2) Generating explanations.

We used HGT as our backbone prediction model. During the training stage, it can effectively converge and achieve solid performance (Macro-F1=99.42%, Micro-F1=99.43%) and (Macro-F1=80.06%, Micro-F1=80.81%) on SynACM and SynDBLP, respectively. During the explanation stage, xPath can successfully generate an explanation subgraph with decent accuracy fidelity, and probability fidelity, Facc and Fprob (Yuan et al., 2020): Facc=0.15665, Fprob=0.15297 on SynACM and Facc=0.16935, Fprob=0.08701 on SynDBLP, both presenting quite reasonable scores. The above preliminary results show that our generated synthetic datasets can indeed be used to evaluate HGNN explanation algorithms. This warrants a more complete further exploration in future work.

## J  COMPUTING RESOURCES

In our experiments, GNN learning utilized an NVIDIA RTX 3060, with fitting a GNN on a heterogeneous information network (HIN) taking under an hour. Graph generation algorithms were executed on a CPU (Intel(R) Core(TM) i7-10700 CPU @ 2.90GHz) with 62GB of RAM, with each graph requiring less than an hour to generate.

## K  USE OF LARGE LANGUAGE MODELS

Large Language Models (LLMs) were utilized for language polishing and manuscript editing. The authors independently designed and conducted all technical content, theoretical results, and experiments.

