# OpenReview forum: "SynHING: Synthetic Heterogeneous Information Network Generation for Graph Learning and Explanation"
_ICLR.cc/2026/Conference — ICLR 2026 Conference Withdrawn Submission_

### Official Review · Reviewer_SBmY · 2025-10-19

**Soundness:** 2
**Presentation:** 3
**Contribution:** 2
**Rating:** 4
**Confidence:** 2

**Summary:**

This paper presents SynHING, a synthetic HIN generation framework that supports both graph learning and explainability research.
To construct synthetic graphs, SynHING first extracts motifs from reference networks, and then recursively combines them based on rules from prior knowledge to form the final HIN. Experiments on IMDB, Recipe, ACM, and DBLP demonstrate that SynHING provides a principled testbed for evaluating Heterogeneous GNNs (HGNNs) and explanation methods.

**Strengths:**

- SynHING is the first framework to enable user-defined, motif-aware HIN synthesis.
- The idea of synthesizing HINs with reference to real HINs is very interesting.

**Weaknesses:**

1. The method used to construct synthetic datasets in this paper is very similar to and straightforward as that for synthesizing homogeneous graphs. Both are based on motif/node labels and handcrafted rules to compose motifs into large graphs.
This work seems to merely replace the motifs of homogeneous graphs with those derived from heterogeneous information networks.

2. The paper defines "target node pairs + all valid meta-paths within a user-defined hop count" as Major Motifs and regards them as "natural explanatory units". However, it lacks key qualitative analysis. For example, measuring the extent to which these motifs can recover the predictions of a GNN trained on the original graph. This is critical because it determines whether the synthesized graphs capture the causal structure of the original graph.

3. Beyond the definition of Major Motifs, the paper also introduces other priors, such as simulating "Superstars" to merge subgraphs. Does this approach make the synthesized graphs closer to the distribution of real graphs? If not, extracting motifs from real HINs is of little significance. Has the author considered designing experiments from the perspective of graph isomorphism analysis or frequent subgraph mining? Has the author considered theoretical analysis to demonstrate the necessity of the proposed synthesis method?

4. In the past, synthesizing graph data has relied on researcher-defined motifs. Thus, the "user-defined motif integration" mentioned in the paper is not a new concept. Additionally, the paper mentions "semantic consistency" multiple times but provides no direct experimental support, there is no direct evidence that the improvement results in Section 5.4 stem from semantic consistency.

In summary, although the paper claims that SynHING is the first framework to enable user-defined, motif-aware HIN synthesis, it fails to address key challenges inherent in this task. These challenges include how to extract causal motifs and how to ensure the synthesis of HINs that are consistent with real distributions. Addressing these challenges is crucial for benchmarking graph learning and explainability research on HINs.

**Questions:**

Please refer to the above weakness section for suggestions and questions.

---

### Official Review · Reviewer_Nvmk · 2025-10-23

**Soundness:** 3
**Presentation:** 2
**Contribution:** 2
**Rating:** 4
**Confidence:** 4

**Summary:**

The paper presents SynHING, a framework for generating synthetic heterogeneous information networks (HINs) with explicit motif-based ground truth that can be used both for explainability benchmarking and controlled model evaluation. The method first defines or extracts key motifs from real datasets, replicates them with controlled noise to form base subgraphs, and then fuses them using a merge operation that preserves edge-type semantics while allowing both intra- and inter-cluster connections. Node features are generated from Gaussian distributions with tunable noise, and minor nodes are added to increase heterogeneity. Two parameters, p and q, control intra- and inter-cluster merging probabilities, enabling precise manipulation of the structural separability and complexity of the generated HINs. Experiments on datasets such as IMDB, Recipe, ACM, and DBLP show that SynHING produces consistent performance trends across heterogeneous GNNs, supports positive transfer when pretraining aligns with real data, and scales linearly up to millions of motifs.

**Strengths:**

- Practical and well-motivated contribution: addresses the lack of motif-aware synthetic benchmarks for heterogeneous GNNs with built-in explainability ground truth.
- Interpretable control knobs: parameters (p, q, SNR) provide fine-grained control over structural purity, homophily, and feature realism; resulting trends in Macro-F1 and explainability fidelity are consistent and interpretable.
- Comprehensive ablations: Random-Motif and Random-Merge baselines demonstrate large performance drops, validating the design choices.
- Demonstrated transfer learning benefit: pretraining on SynHING improves downstream performance on real datasets; corrupted variants induce negative transfer, confirming the framework’s diagnostic value.
- Scalability: theoretical and empirical runtime grows linearly with motif count up to millions of motifs.
- Reproducibility: code and data are available; explainability metric (fidelity) is clearly defined.

**Weaknesses:**

- Fidelity vs. Faithfulness: the paper evaluates explanation quality using fidelity, which measures consistency with the model’s predictions. However, since SynHING provides true causal motifs, the more appropriate metric would be faithfulness,  assessing whether the explanation aligns with the actual causal subgraph [1].
Using both metrics jointly (fidelity + faithfulness) would reveal whether models genuinely learn the motif semantics or merely correlate with spurious structures.
- Missing comparisons: There is no comparison with existing heterogeneous graph generators such as [2, 3]
- Simplistic and potentially biased feature generation:
The feature generation process links node classes to clusters, and hence to distinct feature distributions.
This could make the classification task artificially easier, since each class may be trivially separable by distributional differences rather than by structural cues.
Moreover, in real HINs, different node types typically exhibit distinct feature distributions. The generation process could be improved by sampling distributions per node type.
- Merging operation may introduce illegal edges:
During merging, one node is removed and its neighbors are connected to the remaining node.
However, since edge types encode specific semantic relations, this procedure could create illegal cross-type connections (i.e., edges violating the allowed relation schema).
The paper should clarify whether such type constraints are enforced or whether semantic violations can occur .

[1] Azzolin, S. et al., Reconsidering Faithfulness in Regular, Self-Explainable and Domain Invariant GNNs, ICLR 2025

[2] Ghosh et al., Heterogeneous Graph Generation: A Hierarchical Approach using Node Feature Pooling, 2024

[3] Ling et al., Motif-guided heterogeneous graph deep generation, 2023

**Questions:**

- Since you use fidelity, could you also report faithfulness (explanation–ground truth alignment) to better assess whether the model actually learns the causal motifs?
- Since each class has its own feature distribution, doesn’t this make classification too easy? Would it be more realistic to derive per-node-type distributions from real datasets and use them during generation?
- Could the merge operation create illegal relations across node types? How are relation-type constraints enforced in practice?
- Beyond degree distribution (CDD), have you measured heterogeneous clustering, meta-path distributions, or type assortativity to quantify structural fidelity?
- Does it make sense to include experiments with Multi Relational Graph Neural networks that gives an explanation [4, 5] to check how such models work in this setting? Or at least is this a possible future work that should be included in the related section?
- How does SynHING compare with existing heterogeneous graph generators ?

[4] Ferrini, F. et al., Meta-Path Learning for Multi-Relational Graph Neural Networks, LOG 2023

[5] Ferrini, F. et al, A Self-Explainable Heterogeneous GNN for Relational Deep Learning, TMLR 2025

---

### Official Review · Reviewer_Dd5Q · 2025-10-31

**Soundness:** 1
**Presentation:** 2
**Contribution:** 1
**Rating:** 2
**Confidence:** 4

**Summary:**

This paper introduces a synthetic heterogeneous information network generation framework designed to support both graph learning and explainability research. While the topic is timely and the proposed modular pipeline is conceptually interesting, the work suffers from unclear motivation, insufficient methodological grounding, and weak experimental validation. The idea of user-defined motif-aware graph synthesis is interesting. However, the current presentation lacks rigor and empirical depth to substantiate the claimed contributions.

**Strengths:**

S1. The idea of user-defined motif integration for controllable HIN synthesis is interesting and expands the scope of synthetic graph benchmarks to heterogeneous domains.

S2. The framework is organized with distinct modules and provides mathematical definitions and complexity analysis. Experiments on several real-world HINs have been conducted.

S3. Synthetic datasets are essential for explainability studies. Extending them to heterogeneous graphs could benefit reproducible evaluation of HGNNs.

**Weaknesses:**

W1. The paper doesn’t clearly justify why explainable subgraph or motif discovery in HINs is scientifically necessary. It assumes interpretability is needed without explaining what unique challenges HINs pose compared with homogeneous graphs (e.g., semantic multi-typed relationships, meta-path reasoning). This weakens the conceptual grounding (and motivation) of the work.

W2. The method description needs more clarity. For instance, In Section 3.4, the notion of “randomness injection” and the definition of “minor nodes” are vague. It is unclear what kind of randomness is applied or how it interacts with the added non-target nodes. Figure 1 doesn’t visually distinguish target vs. minor nodes. The “Merge” operation (Section 3.5) is introduced without sufficient theoretical or empirical justification. The motivation behind separating Intra-Cluster Merge and Inter-Cluster Merge is unclear, and their relation to real-world HIN generative processes is not explained.

W3. Experimental design needs to be enhanced. The paper provides no comparison with existing synthetic graph generators (e.g., SBM, GraphWorld, HeteroGraphGAN, etc). Section 5.1 doesn’t have a clear purpose. The results in Figure 4 only show how Macro-F1 varies with parameters 𝑝 and 𝑞, but it is not clear what scientific insight this provides or how it validates the method. The visualizations in Figure 4 (c-e) could be improved. Fidelity is used as a proxy for explainability quality, but no actual explainer or motif-label ground truth is evaluated. Heterogeneous graphs rarely have explicit motif labels, and the authors do not demonstrate that their extracted motifs are semantically meaningful or explanatory.

**Questions:**

Q1. What is the scientific motivation for explainable subgraph or motif mining in heterogeneous information networks, and how SynHING’s goal differs from existing homogeneous graph benchmarks?

Q2. What is “randomness injection”? How are “minor nodes” defined? What “illegal connections” does the Merge operation aim to prevent? Please also explain why both Intra- and Inter-Cluster merges are required and how they relate to real HIN structures.

Q3. Is the post-pruning step manual or automatic? If it is “priority-aware,” what determines the pruning priority and how do users control it?

Q4. How to verify that the extracted motifs are meaningful or explanatory? Are there any quantitative or qualitative analyses to support the claim?

---

### Official Review · Reviewer_owNf · 2025-11-01

**Soundness:** 3
**Presentation:** 3
**Contribution:** 3
**Rating:** 4
**Confidence:** 4

**Summary:**

The paper introduces SynHING, a novel framework for generating synthetic heterogeneous information networks with ground-truth motifs, addressing a critical gap in GNN explainability research. Multiple experiments validate the effectiveness of the synthetic heterogeneous graphs.

**Strengths:**

1. This paper proposes a novel method for synthesizing heterogeneous graph datasets. This approach is promising since the field of heterogeneous graph explainers lacks high-quality synthetic datasets.
2. The authors provide explanations and detailed experiments to demonstrate their method.
3. The paper is easy to follow, with the main components.

**Weaknesses:**

1. The paper highlights the ability to use "custom motifs". Could you clarify the structural limitations on these motifs? For example, must they adhere to the "two anchor nodes" structure described in the MMG module, or could the framework integrate more complex, arbitrary graphlets (e.g., a 5-clique, a star-graph centered on a minor node) as the core explanatory unit?
2. As a new dataset method, its effectiveness needs to be verified on more heterogeneous graph neural networks. At the same time, the author lacks in-depth demonstration of the experimental results. For example, in the experimental results in Table 2, this may be because the target node features obtained by sampling the proposed SynHING are the result of enhancing the original features.
3. The method proposed in the paper is suitable for node classification tasks, but lacks experiments on other tasks.
4. Figure 1 and figure 2 are too small, and they lack relevant detailed captions, making them difficult to read.
5. The paper focuses on fidelity (sufficiency) and mentions explainer support, but a richer metric could be included, such as infidelity, sparsity, and AUC.

**Questions:**

Please see weaknesses.

---

### Note · Authors · 2025-11-21

**Comment:**

We sincerely thank all the reviewers for their detailed and constructive reviews. We have decided to withdraw this submission and thoroughly revise it by incorporating all the suggestions and addressing all the concerns and questions. We appreciate the reviewers' tremendous effort in helping us improve our submission.

**Withdrawal Confirmation:**

I have read and agree with the venue's withdrawal policy on behalf of myself and my co-authors.